# Smooth Exploration for Robotic Reinforcement Learning

**Antonin Raffin**
Robotics and Mechatronics Center (RMC)
German Aerospace Center (DLR) Germany
`antonin.raffin@dlr.de`

**Jens Kober**
Cognitive Robotics Department
Delft University of Technology The Netherlands
`j.kober@tudelft.nl`

**Freek Stulp**
Robotics and Mechatronics Center (RMC)
German Aerospace Center (DLR) Germany
`freek.stulp@dlr.de`

**Abstract:**

Reinforcement learning (RL) enables robots to learn skills from interactions with the real world. In practice, the unstructured step-based exploration used in Deep RL – often very successful in simulation – leads to jerky motion patterns on real robots. Consequences of the resulting shaky behavior are poor exploration, or even damage to the robot. We address these issues by adapting state-dependent exploration (SDE) [1] to current Deep RL algorithms. To enable this adaptation, we propose two extensions to the original SDE, using more general features and re-sampling the noise periodically, which leads to a new exploration method *generalized state-dependent exploration* (*g*SDE). We evaluate *g*SDE both in simulation, on PyBullet continuous control tasks, and directly on three different real robots: a tendon-driven elastic robot, a quadruped and an RC car. The noise sampling interval of *g*SDE enables a compromise between performance and smoothness, which allows training directly on the real robots without loss of performance.

## 1 Introduction

One of the first robots that used artificial intelligence methods was called "Shakey", because it would shake a lot during operation [2]. Shaking has now again become quite prevalent in robotics, but for a different reason. When learning robotic skills with deep reinforcement learning (DeepRL), the de facto standard for exploration is to sample a noise vector $\epsilon_t$ from a Gaussian distribution independently at each time step $t$, and then adding it to the policy output. This approach leads to the type of noise illustrated to the left in Fig. 1, and it can be very effective in simulation [3, 4, 5, 6, 7].

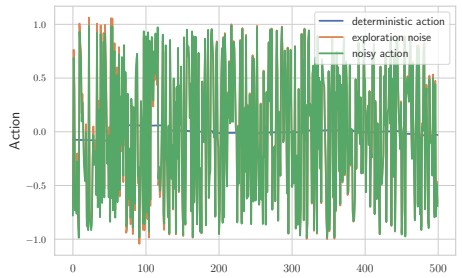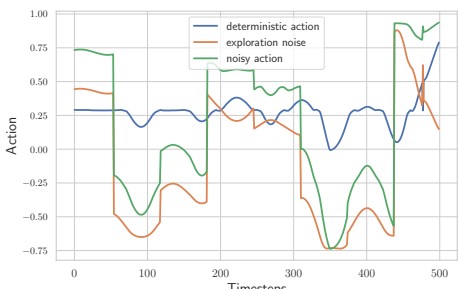

Figure 1: Left: unstructured exploration, as typically used in simulated RL. Right: *g*SDE provides smooth and consistent exploration.

5th Conference on Robot Learning (CoRL 2021), London, UK.

Unstructured exploration has also been applied to robotics [8, 9]. But for experiments on real robots, it has many drawbacks, which have been repeatedly pointed out [1, 10, 11, 12, 13]: 1) Sampling independently at each step leads to shaky behavior [14], and noisy, jittery trajectories. 2) The jerky motion patterns can damage the motors on a real robot, and lead to increased wear-and-tear. 3) In the real world, the system acts as a low pass filter. Thus, consecutive perturbations may cancel each other, leading to poor exploration. This is particularly true for high control frequency [15]. 4) It causes a large variance which grows with the number of time-steps [10, 11, 12]

In practice, we have observed all of these drawbacks on three real robots, including the tendon-driven robot David, depicted in Fig. 4a, which is the main experimental platform used in this work. For all practical purposes, Deep RL with unstructured noise cannot be applied to David.

In robotics, multiple solutions have been proposed to counteract the inefficiency of unstructured noise. These include correlated noise [8, 15], low-pass filters [16, 17], action repeat [18] or lower level controllers [16, 9]. A more principled solution is to perform exploration in parameter space, rather than in action space [19, 20]. This approach usually requires fundamental changes in the algorithm, and is harder to tune when the number of parameters is high.

State-Dependent Exploration (SDE) [1, 11] was proposed as a compromise between exploring in parameter and action space. SDE replaces the sampled noise with a state-dependent exploration function, which during an episode returns the same action for a given state. This results in smoother exploration and less variance per episode.

To the best of our knowledge, no Deep RL algorithm has yet been successfully combined with SDE. We surmise that this is because the problem that it solves – shaky, jerky movement – is not as noticeable in simulation, which is the current focus of the community.

In this paper, we aim at reviving interest in SDE as an effective method for addressing exploration issues that arise from using independently sampled Gaussian noise on real robots. Our concrete contributions, which also determine the structure of the paper, are:

1. Highlighting the issues with unstructured Gaussian exploration (Sect. 1).
2. Adapting SDE to recent Deep RL algorithms, and addressing some issues of the original formulation (Sects. 2.2 and 3).
3. Evaluate the different approaches with respect to the compromise between smoothness and performance, and show the impact of the noise sampling interval (Sects. 4.1 and 4.2).
4. Successfully applying RL directly on three real robots: a tendon-driven robot, a quadruped and an RC car, without need of a simulator or filters (Sect. 4.3).

## 2 Background

In reinforcement learning, an agent interacts with its environment, usually modeled as a Markov Decision Process (MDP) $(\mathcal{S}, \mathcal{A}, p, r)$ where $\mathcal{S}$ is the state space, $\mathcal{A}$ the action space and $p(\mathbf{s}'|\mathbf{s}, \mathbf{a})$ the transition function. At every step $t$, the agent performs an action $\mathbf{a}$ in state $\mathbf{s}$ following its policy $\pi : \mathcal{S} \mapsto \mathcal{A}$. It then receives a feedback signal in the next state $\mathbf{s}'$: the reward $r(\mathbf{s}, \mathbf{a})$. The objective of the agent is to maximize the long-term reward. More formally, the goal is to maximize the expectation of the sum of discounted reward, over the trajectories $\rho_\pi$ generated using its policy $\pi$:

$$\sum_t \mathbb{E}_{(\mathbf{s}_t, \mathbf{a}_t) \sim \rho_\pi} \left[ \gamma^t r(\mathbf{s}_t, \mathbf{a}_t) \right] \tag{1}$$

where $\gamma \in [0, 1)$ is the discount factor and represents a trade-off between maximizing short-term and long-term rewards. The agent-environment interactions are often broken down into sequences called *episodes*, that end when the agent reaches a terminal state.

### 2.1 Exploration in Action or Policy Parameter Space

In the case of continuous actions, the exploration is commonly done in the *action space* [21, 22, 23, 24, 25, 5]. At each time-step, a noise vector $\epsilon_t$ is independently sampled from a Gaussian distribution and then added to the controller output:

$$\mathbf{a}_t = \mu(\mathbf{s}_t; \theta_\mu) + \epsilon_t, \qquad \qquad \epsilon_t \sim \mathcal{N}(0, \sigma^2) \tag{2}$$

where $\mu(\mathbf{s}_t)$ is the deterministic policy and $\pi(\mathbf{a}_t|\mathbf{s}_t) \sim \mathcal{N}(\mu(\mathbf{s}_t), \sigma^2)$ is the resulting stochastic policy, used for exploration. $\theta_\mu$ denotes the parameters of the deterministic policy.

For simplicity, throughout the paper, we will only consider Gaussian distributions with diagonal covariance matrices. Hence, here, $\sigma$ is a vector with the same dimension as the action space $\mathcal{A}$.

Alternatively, the exploration can also be done in the *parameter space* [11, 19, 20]:

$$\mathbf{a}_t = \mu(\mathbf{s}_t; \theta_\mu + \epsilon), \qquad\qquad \epsilon \sim \mathcal{N}(0, \sigma^2) \tag{3}$$

at the beginning of an episode, the perturbation $\epsilon$ is sampled and added to the policy parameters $\theta_\mu$. This usually results in more consistent exploration but becomes challenging with an increasing number of parameters [19].

## 2.2 State-Dependent Exploration

*State-Dependent Exploration (*SDE*)* [1, 11] is an intermediate solution that consists in adding noise as a function of the state $\mathbf{s}_t$, to the deterministic action $\mu(\mathbf{s}_t)$. At the beginning of an episode, the parameters $\theta_\epsilon$ of that exploration function are drawn from a Gaussian distribution. The resulting action $\mathbf{a}_t$ is as follows:

$$\mathbf{a}_t = \mu(\mathbf{s}_t; \theta_\mu) + \epsilon(\mathbf{s}_t; \theta_\epsilon), \qquad\qquad \theta_\epsilon \sim \mathcal{N}(0, \sigma^2) \tag{4}$$

This episode-based exploration is smoother and more consistent than the unstructured step-based exploration. Thus, during one episode, instead of oscillating around a mean value, the action $\mathbf{a}$ for a given state $\mathbf{s}$ will be the same.

SDE should not be confused with unstructured noise where the variance can be state-dependent but the noise is still sampled at every step, as it is the case for SAC.

In the remainder of this paper, to avoid overloading notation, we drop the time subscript $t$, i.e. we now write $\mathbf{s}$ instead of $\mathbf{s}_t$. $\mathbf{s}_j$ or $\mathbf{a}_j$ now refer to an element of the state or action vector.

In the case of a linear exploration function $\epsilon(\mathbf{s}; \theta_\epsilon) = \theta_\epsilon \mathbf{s}$, by operation on Gaussian distributions, Rückstieß et al. [1] show that the action element $\mathbf{a}_j$ is normally distributed:

$$\pi_j(\mathbf{a}_j|\mathbf{s}) \sim \mathcal{N}(\mu_j(\mathbf{s}), \hat{\sigma_j}^2) \tag{5}$$

where $\hat{\sigma}$ is a diagonal matrix with elements $\hat{\sigma}_j = \sqrt{\sum_i (\sigma_{ij} \mathbf{s}_i)^2}$.

Because we know the policy distribution, we can obtain the derivative of the log-likelihood $\log \pi(\mathbf{a}|\mathbf{s})$ with respect to the variance $\sigma$:

$$\frac{\partial \log \pi(\mathbf{a}|\mathbf{s})}{\partial \sigma_{ij}} = \frac{(\mathbf{a}_j - \mu_j)^2 - \hat{\sigma_j}^2}{\hat{\sigma_j}^3} \frac{\mathbf{s}_i^2 \sigma_{ij}}{\hat{\sigma_j}} \tag{6}$$

This can be easily plugged into the likelihood ratio gradient estimator [26], which enables adaptation of $\sigma$ during training. SDE is therefore compatible with standard policy gradient methods, while addressing most shortcomings of the unstructured exploration.

For a non-linear exploration function, the resulting distribution $\pi(\mathbf{a}|\mathbf{s})$ is most of the time unknown. Thus, computing the exact derivative w.r.t. the variance is not trivial and may require approximate inference. As we focus on simplicity, we leave this extension for future work.

## 3 Generalized State-Dependent Exploration

Considering Eqs. (5) and (6), some limitations of the original formulation are apparent:

    i The noise does not change during one episode, which is problematic if the episode length is long, because the exploration will be limited [27].

    ii The variance of the policy $\hat{\sigma}_j = \sqrt{\sum_i (\sigma_{ij} \mathbf{s}_i)^2}$ depends on the state space dimension (it grows with it), which means that the initial $\sigma$ must be tuned for each problem.

    iii There is only a linear dependency between the state and the exploration noise, which limits the possibilities.

iv The state must be normalized, as the gradient and the noise magnitude depend on the state magnitude.

To mitigate the mentioned issues and adapt it to Deep RL algorithms, we propose two improvements:

1. We sample the parameters $\theta_\epsilon$ of the exploration function every $n$ steps instead of every episode.

2. Instead of the state $\mathbf{s}$, we can in fact use any features. We chose policy features $\mathbf{z}_\mu(\mathbf{s}; \theta_{\mathbf{z}_\mu})$ (last layer before the deterministic output $\mu(\mathbf{s}) = \theta_\mu \mathbf{z}_\mu(\mathbf{s}; \theta_{\mathbf{z}_\mu})$) as input to the noise function $\epsilon(\mathbf{s}; \theta_\epsilon) = \theta_\epsilon \mathbf{z}_\mu(\mathbf{s})$.

Sampling the parameters $\theta_\epsilon$ every $n$ steps tackles the issue i. and yields a unifying framework [27] which encompasses both unstructured exploration ($n = 1$) and original SDE ($n = \mathrm{episode\_length}$). Although this formulation follows the description of Deep RL algorithms that update their parameters every $m$ steps, the influence of this crucial parameter on smoothness and performance was until now overlooked.

Using *policy features* allows mitigating issues ii, iii and iv: the relationship between the state $\mathbf{s}$ and the noise $\epsilon$ is non-linear and the variance of the policy only depends on the network architecture. This makes $g$SDE more task-independent (as the network architecture is usually kept constant) and saves a lot of parameters and computation when working with large state space such as images: the number of parameters and operations is only a function of the last layer size and action dimension and no more of the state space size. This formulation is therefore more general and includes the original SDE description, when using state as input to the noise function or when the policy is linear.

We call the resulting approach *generalized* State-Dependent Exploration ($g$SDE).

**Deep RL algorithms**  Integrating this updated version of SDE into recent Deep RL algorithms, such as those listed in the appendix, is straightforward. For those that rely on a probability distribution, such as SAC or PPO, we can replace the original Gaussian distribution by the one from Eq. (5), where the analytical form of the log-likelihood is known (cf. Eq. (6)).

## 4    Experiments

In this section, we study $g$SDE to answer the following questions:

- How does $g$SDE compares to the original SDE? What is the impact of each proposed modification?

- How does $g$SDE compares to other type of exploration noise in terms of compromise between smoothness and performance?

- How does $g$SDE performs on a real system?

### 4.1    Compromise Between Smoothness and Performance

**Experiment setup**   In order to compare $g$SDE to other type of exploration in terms of compromise between performance and smoothness, we chose 4 locomotion tasks from the PyBullet [28] environments: HALFCHEETAH, ANT, HOPPER and WALKER2D. They are similar to the one found in OpenAI Gym [29] but the simulator is open source and they are harder to solve[1]. In this section, we focus on the SAC algorithm as it will be the one used on the real robot, although we report results for additional algorithms such as PPO in the appendix.

To evaluate smoothness, we define a continuity cost $\mathcal{C} = 100 \times \mathbb{E}_t \left[ \left( \frac{\mathbf{a}_{t+1} - \mathbf{a}_t}{\Delta^{\mathbf{a}}_{\max}} \right)^2 \right]$ which yields values between 0 (constant output) and 100 (action jumping from one limit to another at every step). The continuity cost of the training $\mathcal{C}_{\mathrm{train}}$ is a proxy for the wear-and-tear of the robot.

---

[1] https://frama.link/PyBullet-harder-than-MuJoCo

We compare the performance of the following configurations: (a) no exploration noise, (b) unstructured Gaussian noise (original SAC implementation), (c) correlated noise (Ornstein–Uhlenbeck process [30] with $\sigma=0.2$, OU noise in the figure), (d) adaptive parameter noise [19] ($\sigma=0.2$), (e) $g$SDE. To decorrelate the exploration noise from the one due to parameter update, and to be closer to a real robot setting, we apply the gradient updates only at the end of each episode.

We fix the budget to 1 million steps and report the average score over 10 runs together with the average continuity cost during training and their standard error. For each run, we test the learned policy on 20 evaluation episodes every 10000 steps, using the deterministic controller $\mu(\mathbf{s}_t)$. Regarding the implementation, we use a modified version of Stable-Baselines3 [31] together with the RL Zoo training framework [32]. The methodology we follow to tune the hyperparameters and their details can be found in the appendix. The code we used to run the experiments and tune the hyperparameters can be found in the supplementary material.

| Algorithm | HALFCHEETAH | | ANT | | HOPPER | | WALKER2D | |
|---|---|---|---|---|---|---|---|---|
| SAC | Return ↑ | $\mathcal{C}_{\text{train}}$ ↓ | Return ↑ | $\mathcal{C}_{\text{train}}$ ↓ | Return ↑ | $\mathcal{C}_{\text{train}}$ ↓ | Return ↑ | $\mathcal{C}_{\text{train}}$ ↓ |
| w/o noise | 2562 +/- 102 | **2.6** +/- 0.1 | 2600 +/- 364 | **2.0** +/- 0.2 | 1661 +/- 270 | **1.8** +/- 0.1 | 2216 +/- 40 | **1.8** +/- 0.1 |
| w/ unstructured | **2994** +/- 89 | 4.8 +/- 0.2 | **3394** +/- 64 | 5.1 +/- 0.1 | **2434** +/- 190 | 3.6 +/- 0.1 | 2225 +/- 35 | 3.6 +/- 0.1 |
| w/ OU noise | 2692 +/- 68 | 2.9 +/- 0.1 | 2849 +/- 267 | 2.3 +/- 0.0 | 2200 +/- 53 | 2.1 +/- 0.1 | 2089 +/- 25 | 2.0 +/- 0.0 |
| w/ param noise | 2834 +/- 54 | 2.9 +/- 0.1 | 3294 +/- 55 | **2.1** +/- 0.1 | 1685 +/- 279 | 2.2 +/- 0.1 | **2294** +/- 40 | **1.8** +/- 0.1 |
| w/ $g$SDE-2 | **2987** +/- 85 | 4.1 +/- 0.2 | 3366 +/- 50 | 4.7 +/- 0.1 | **2532** +/- 70 | 2.8 +/- 0.1 | 2237 +/- 55 | 2.8 +/- 0.1 |
| w/ $g$SDE-4 | 2798 +/- 41 | 4.1 +/- 0.2 | 3227 +/- 182 | 3.8 +/- 0.2 | 2541 +/- 49 | 2.6 +/- 0.1 | **2322** +/- 69 | 2.6 +/- 0.1 |
| w/ $g$SDE-8 | 2850 +/- 73 | 4.1 +/- 0.2 | **3459** +/- 52 | 3.9 +/- 0.2 | **2646** +/- 45 | 2.4 +/- 0.1 | **2341** +/- 45 | 2.5 +/- 0.1 |
| w/ $g$SDE-64 | **2970** +/- 132 | 3.5 +/- 0.1 | 3160 +/- 184 | 3.5 +/- 0.1 | 2476 +/- 99 | **2.0** +/- 0.1 | **2324** +/- 39 | 2.3 +/- 0.1 |
| w/ $g$SDE-Episodic | 2741 +/- 115 | 3.1 +/- 0.2 | 3044 +/- 106 | 2.6 +/- 0.1 | 2503 +/- 80 | **1.8** +/- 0.1 | 2267 +/- 34 | 2.2 +/- 0.1 |

Table 1: Detailed return and continuity cost results for SAC with different type of exploration on PyBullet environments. We report the mean and standard error over 10 runs of 1 million steps. For each benchmark, we highlight the results of the method(s) with the best mean when the difference is statistically significant.

**Results** Table 1 and Fig. 2 shows the results on the PyBullet tasks and the compromise between continuity and performance. Without any noise ("No Noise" in the figure), SAC is still able to solve partially those tasks thanks to a shaped reward, but it has the highest variance in the results. Although the correlated and parameter noise yield lower continuity cost during training, it comes at a cost of performance. $g$SDE is able to achieve a good compromise between unstructured exploration and correlated noise by making use of the noise repeat parameter. $g$SDE-8 (sampling the noise every 8 steps) even achieves better performance with a lower continuity cost at train time. Such behavior is what is desirable for training on a real robot: we must minimize wear-and-tear at training while still obtaining good performance at test time.

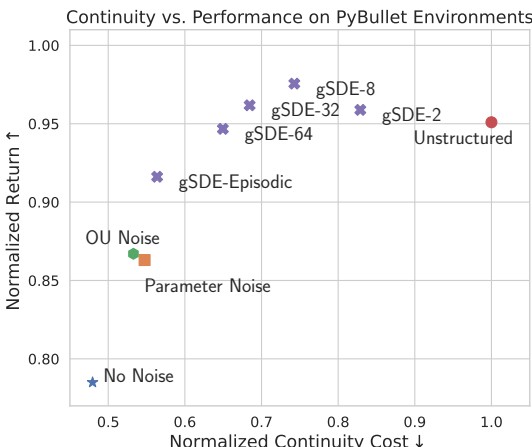

Figure 2: Normalized return and continuity cost of SAC on 4 PyBullet tasks with different type of exploration. $g$SDE provides a compromise between performance and smoothness.

## 4.2 Comparison to the Original SDE

In this section, we investigate the contribution of the proposed modifications to the original SDE: sampling the exploration function parameters every $n$ steps and using policy features as input to the noise function.

**Sampling Interval** $g$SDE is a $n$-step version of SDE, where $n$ allows interpolating between the unstructured exploration $n = 1$ and the original SDE per-episode formulation. This interpolation permits a compromise between performance and smoothness at train time (cf. Table 1 and Fig. 2). Fig. 3b shows the importance of that parameter for PPO on the WALKER2D task. If the sampling interval is too large, the agent would not explore enough during long episodes. On the other hand, with a high sampling frequency $n \approx 1$, the issues mentioned in Sect. 1 arise.

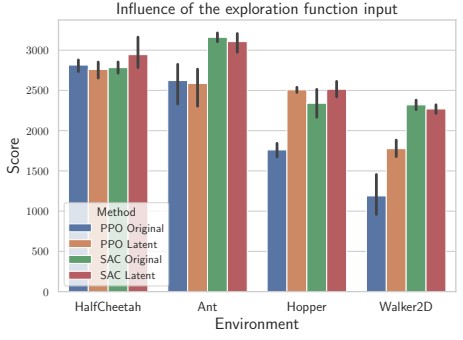

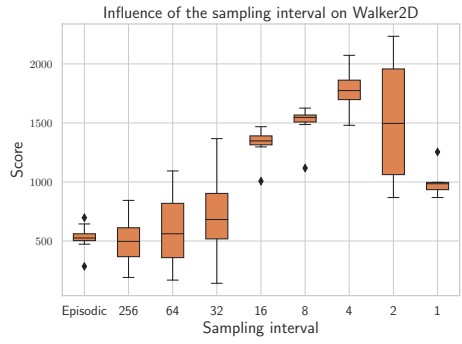

(a) Exploration function input

(b) Sampling interval (PPO on WALKER2D)

Figure 3: Impact of $g$SDE modifications over the original SDE on PyBullet tasks. (a) Influence of the input to the exploration function $\epsilon(\mathbf{s}; \theta_\epsilon)$ for SAC and PPO: using latent features from the policy $\mathbf{z}_\mu$ (Latent) is usually better than using the state $\mathbf{s}$ (Original). (b) The frequency of sampling the noise function parameters is crucial for PPO with $g$SDE.

**Policy features as input**   Fig. 3a shows the effect of changing the exploration function input for SAC and PPO. Although it varies from task to task, using policy features ("latent" in the figure) is usually beneficial, especially for PPO. It also requires less tuning and no normalization as it depends only on the policy network architecture. Here, the PyBullet tasks are low dimensional and the state space size is of the same order, so no careful per-task tuning is needed. Relying on features also allows learning directly from pixels, which is not possible in the original formulation.

Compared to the original SDE, the two proposed modifications are beneficial to the performance, with the noise sampling interval $n$ having the most impact. Fortunately, as shown in Table 1 and Fig. 2, it can be chosen quite freely for SAC. In the appendix, we provide an additional ablation study that shows $g$SDE is robust to the choice of the initial exploration variance.

### 4.3   Learning to Control a Tendon-Driven Elastic Robot

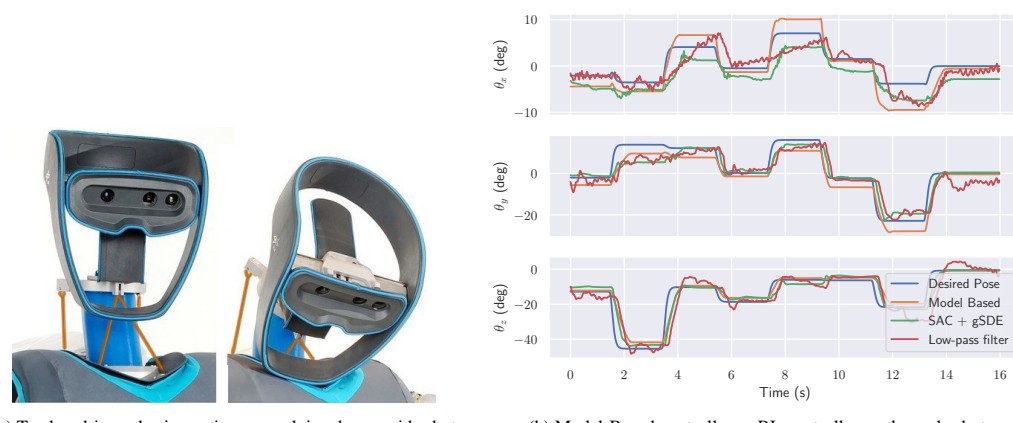

(a) Tendon-driven elastic continuum neck in a humanoid robot

(b) Model-Based controller vs RL controller on the real robot

Figure 4: (a) The tendon-driven robot [33] used for the experiment. The tendons are highlighted in orange. (b) The model-based controller and the RL agent performs similarly on an evaluation trajectory.

**Experiment setup**   To assess the usefulness of $g$SDE, we apply it on a real system. The task is to control a tendon-driven elastic continuum neck [33] (see Fig. 4a) to a given target pose. Controlling such a soft robot is challenging because of the nonlinear tendon coupling, together with the deformation of the structure that needs to be modeled accurately. This modeling is computationally expensive [34, 35] and requires assumptions that may not hold in the physical system.

The system is under-actuated (there are only 4 tendons), hence, the desired pose is a 4D vector: 3 angles for the rotation $\theta_x$, $\theta_y$, $\theta_z$ and one for the position $x$. The input is a 16D vector composed of: the measured tendon lengths (4D), the current tendon forces (4D), the current pose (4D) and the target pose (4D). The reward is a weighted sum between the negative geodesic distance to the desired orientation and the negative Euclidean distance to the desired position. The weights are chosen such that the two components have the same magnitude. We also add a small continuity cost to reduce the oscillations in the final policy. The action space consists in desired delta in tendon forces, limited to $5\,\mathrm{N}$. For safety reasons, the tendon forces are clipped below $10\,\mathrm{N}$ and above $40\,\mathrm{N}$. An episode terminates either when the agent reaches the desired pose or after a timeout of $5\,\mathrm{s}$. The episode is considered successful if the desired pose is reached within a threshold of $10\,\mathrm{mm}$ for the position and $5°$ for the orientation. The agent controls the tendons forces at $30\,\mathrm{Hz}$, while a PD controller monitors the motor current at $3\,\mathrm{kHz}$ on the robot. Gradient updates are directly done on a 4-core laptop, after each episode.

**Results**  We first ran the unstructured exploration on the robot, but had to stop the experiment early: the high-frequency noise in the command was damaging the tendons and would have broken them due to their friction on the bearings. Therefore, as a baseline, we trained a policy using SAC with a hand-crafted action smoothing ($2\,\mathrm{Hz}$ cutoff Butterworth low-pass filter) for two hours.

Then, we trained a controller using SAC with $g$SDE for the same duration. We compare both learned controllers to an existing model-based controller (passivity-based approach) presented in [34, 35] using a pre-defined trajectory (cf. Fig. 4b). On the evaluation trajectory, the controllers are equally precise (cf. Table 2): the mean error in orientation is below $3°$ and the one in position below $3\,\mathrm{mm}$. However, the policy trained with the low-pass filter is much more jittery than the two others. We quantify this jitter as the mean absolute difference between two timesteps, denoted as *continuity cost* in Table 2.

|  | Unstructured noise | $g$SDE | Low-pass filter | Model-Based |
|---|---|---|---|---|
| Position error (mm) | N/A | 2.65 +/- 1.6 | 1.98 +/- 1.7 | 1.32 +/- 1.2 |
| Orientation error (deg) | N/A | 2.85 +/- 2.9 | 3.53 +/- 4.0 | 2.90 +/- 2.8 |
| Continuity cost (deg) | N/A | **0.20 +/- 0.04** | 0.38 +/- 0.07 | **0.16 +/- 0.04** |

Table 2: Comparison of the mean error in position, orientation and mean continuity cost on the evaluation trajectory. We highlight best approaches when the difference is significant. The model-based and learned controllers yield comparable results but the policy trained with a low-pass filter has a much higher continuity cost.

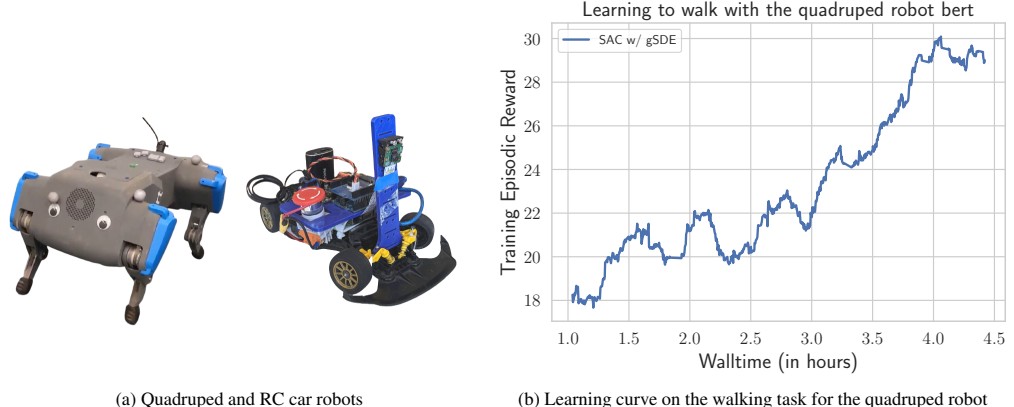

(a) Quadruped and RC car robots        (b) Learning curve on the walking task for the quadruped robot

Figure 5: Additional robots successfully trained using SAC with $g$SDE directly in the real world.

**Additional Real Robot Experiments**  We also successfully applied SAC with $g$SDE on two additional real robotics tasks (see Fig. 5a): training an elastic quadruped robot to walk (learning curve in Fig. 5b) and learning to drive around a track using an RC car. Both experiments are fully done on

the real robot, without the use of simulation nor filter. We provide in the supplementary material the videos of the trained controllers.

# 5   Related Work

Exploration is a key topic in reinforcement learning [36]. It has been extensively studied in the discrete case and most recent papers still focus on discrete actions [37, 38].

Several works tackle the issues of unstructured exploration for continuous control by replacing it with correlated noise. Korenkevych et al. [15] use an autoregressive process and introduce two variables that allows to control the smoothness of the exploration. In the same vein, van Hoof et al. [27] rely on a temporal coherence parameter to interpolate between the step- or episode-based exploration, making use of a Markov chain to correlate the noise. This smoothed noise comes at a cost: it requires an history, which changes the problem definition.

Exploring in parameter space [10, 39, 11, 12, 40] is an orthogonal approach that also solves some issues of the unstructured exploration. It was successfully applied to real robot but relied on motor primitives [41, 12], which requires expert knowledge. Plappert et al. [19] adapt parameter exploration to Deep RL by defining a distance in the action space and applying layer normalization to handle high-dimensional space.

Population based algorithms, such as Evolution strategies (ES) or Genetic Algorithms (GA), also explore in parameter space. Thanks to massive parallelization, they were shown to be competitive [42] with RL in terms of training time, at the cost of being sample inefficient. To address this problem, recent works [20] proposed to combine ES exploration with RL gradient update. This combination, although powerful, unfortunately adds numerous hyperparameters and a non-negligible computational overhead.

Obtaining smooth control is essential for real robot, but it is usually overlooked by the DeepRL community. Recently, Mysore et al. [14] integrated a continuity and smoothing loss inside RL algorithms. Their approach is effective to obtain a smooth controller that reduces the energy used at test time on the real robot. However, it does not solve the issue of smooth exploration at train time, limiting their training to simulation only.

# 6   Conclusion

In this work, we highlighted several issues that arise from the unstructured exploration in Deep RL algorithms for continuous control. Due to those issues, these algorithms cannot be directly applied to learning on real-world robots.

To address these issues, we adapt State-Dependent Exploration to Deep RL algorithms by extending the original formulation: we sample the noise every $n$ steps and replace the exploration function input by learned features. This generalized version ($g$SDE), provides a simple and efficient alternative to unstructured Gaussian exploration.

$g$SDE achieves very competitive results on several continuous control benchmarks, while reducing wear-and-tear at train time. We also investigate the contribution of each modification by performing an ablation study: the noise sampling interval has the most impact and allows a compromise between performance and smoothness. Our proposed exploration strategy, combined with SAC, is robust to hyperparameter choice, which makes it suitable for robotics applications. To demonstrate it, we successfully apply SAC with $g$SDE directly on three different robots.

Although much progress is being made in *sim2real* approaches, we believe more effort should be invested in learning directly on real systems, even if this poses challenges in terms of safety and duration of learning. This paper is meant as a step towards this goal, and we hope that it will revive interest in developing exploration methods that can be directly applied to real robots.

**Acknowledgments**

The work described in this paper was partially funded by the project "Reduced Complexity Models" from the "Helmholtz-Gemeinschaft Deutscher Forschungszentren" and by the EU H2020 project

"VERtical Innovation in the Domain of Robotics Enabled by Artificial intelligence Methods". We would like to thank all the people that gave us valuable feedback on an early version of this paper.

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
