# OpenReview forum: "Smooth Exploration for Robotic Reinforcement Learning"
_robot-learning.org/CoRL/2021/Conference — CoRL2021 Poster_

### Official Review · Reviewer_uHC8 · 2021-07-14

**Originality:** Fair
**Technical Quality:** Very Good
**Clarity Of Presentation:** Very Good
**Impact:** 2

**Recommendation:**

Weak Accept: I recommend accepting the paper, but will not argue for my recommendation if the majority of other reviewers have a different opinion.

**Summary:**

This paper proposes a new method (gSDE) for generating exploration noise for RL, conditioned on the last hidden layer of the policy network and re-generated at a frequency tunable between every timestep and every episode. This is a modification of state-dependent exploration (SDE), with the differences being 1) SDE conditions on the environment state instead of the network, and 2) SDE always re-generates once per episode. gSDE is evaluated on 4 MuJoCo tasks in simulation and 3 real-world systems and found to generate smoother action sequences than standard unstructured Gaussian noise with similar returns. The update frequency can be adjusted to modulate the trade-off between smoothness and performance.


**Issues:**

* Please clarify the reasons for using policy features as exploration function input. I understand that it eliminates the need for normalization if the features are already normalized. I do not understand why SDE cannot use pixel input--wouldn't it simply require a random sample (or 3) per pixel? Is the concern the quantity of samples and multiplications?

**Reviewer Expertise:**

Good: General knowledge of the area

**Strengths And Weaknesses:**

Strengths:
* The problem is very relevant. We now have RL algorithms that are data-efficient enough to allow real-world training, which means hardware damage from noisy actions has become a blocker.
* The MuJoCo tasks, head actuation, and quadruped locomotion results are convincing.

Weaknesses:
* The changes to SDE are very slight.
* In the video of head actuation, the comparison between exploration with per-step noise and with gSDE is useful. It would be nice to have the same comparison for the quadruped.


**Summary Of Recommendation:**

The work is a relatively simple extension of SDE without a huge novelty value. However, the authors correctly point out that SDE is *not* used in RL, and gSDE solves the problems that make SDE less effective. I believe it is worthwhile to demonstrate that this simple method can make real-world training feasible for systems where it otherwise wouldn't be.

---

> ### Author Response · Authors · 2021-08-26
> **Reply to Reviewer uHC8**
>
>
> Dear Reviewer,
>
> Thank you for the insightful feedback!
> Please find below our replies to your comments.
>
> >Please clarify the reasons for using policy features as exploration function input. I understand that it eliminates the need for normalization if the features are already normalized. I do not understand why SDE cannot use pixel input--wouldn't it simply require a random sample (or 3) per pixel? Is the concern the quantity of samples and multiplications?
>
> It does eliminate the need for normalization but additionally, it does not require a per-task tuning (as the network architecture is usually kept constant) and was shown to work well on a good range of tasks.
> SDE on pixel input could be used in theory, but in practice, as Reviewer #uHC8 guessed, it adds many parameters and operations.
>
> As an example, let's consider a task with a 64x64x3 image as input (classic dimensions for Atari games) and 6D action as output.
>
> With SDE, the exploration function would have 64x64x3x6 ~= 12200 x 6  ~= 73k parameters and requires 12200 x 73000 ~= 890M multiplications.
>
> With gSDE, the exploration function would have Hx6 ~= 3k parameters (where H=512 is the last hidden layer size of the DQN Nature CNN architecture, could be smaller in practice) and requires HxHx6 ~= 1.6M multiplications.
>
> As shown by this simple example (it would be true for any large state space), gSDE can drastically reduce the number of parameters and operations (by a factor 100 here!).

---

> > ### Comment · Reviewer_uHC8 · 2021-09-02
> > **Response to authors**
> >
> > Thank you for the clarification, and for the updates to the paper. I stand by my initial recommendation of weak accept--I think this is a promising, easy-to-implement technique, and I look forward to trying it out.

---

### Official Review · Reviewer_8waC · 2021-07-23

**Originality:** Poor
**Technical Quality:** Excellent
**Clarity Of Presentation:** Excellent
**Impact:** 4

**Recommendation:**

Strong Accept: I recommend accepting the paper and will argue for my recommendation even if other reviewers hold a different opinion.

**Summary:**

This work addresses the ubiquitous problem of shaky controllers trained using model free deep RL by proposing the exploration technique, generalized State Dependent Exploration (gSDE). gSDE builds on prior work SDE and substitutes the per-step random Gaussian action noise in many deep RL algorithms with state-dependent noise that only varies every n-steps. The effect is smoother and more consistent exploration trajectories which reduce wear and tear without substantially compromising performance. gSDE is assessed in simulation as well as on three different physical robotic systems.


**Issues:**

Please address the following comments and questions:
* Lines 100 - 126: The claim that gSDE addresses all four of the identified limitations with SDE is not fully convincing
    1. (ii) Now the variance of the policy depends on the policy features dimension so needs to be tuned for each network.
    2. (iv) For similar reasons to those given in (iv) (lines 108 -109) shouldn’t the policy features now be normalized?
* Figure 2: Given the apparent linear relationship between the frequency of noise parameter re-sampling and continuity cost, the sudden drop-off to the Unstructured noise is surprising. It is achieving returns equivalent to gSDE-64 but with almost 2x normalized cost
    1. There is a similar puzzling relationship occurring in Figure 3b)
    2. To better understand this it would be helpful to add gSDE-4 and gSDE-2 to Figure 2 and gSDE 2 to Figure 3b).
    3. I would expect a somewhat smooth transition between gSDE-2 and unstructured noise. Regardless, adding these points would be illuminating.
* Table 2: Is there just one evaluation trajectory?
* Grammar
    1. Line 13: “permits to have” → facilitates / enables?
    2. Line 54: “without the need of a” -> “without need of a” / “without the need for”
    3. Line 94: “allows to adapt” → “enables adaptation of”
    4. Line 184: “allows to have” → “permits a” / “makes it possible to”

**Reviewer Expertise:**

Very good: Comprehensive knowledge of the area

**Strengths And Weaknesses:**

**Strengths**

* Effectively addresses a ubiquitous problem in Deep RL for robotics - shaky controllers.
* The proposed method, gSDE, is widely applicable. It can be applied to any continuous control problem that is solved using either a pure policy gradient method (e.g. REINFORCE) or a hybrid policy and value based method (e.g. PPO, A2C, TD3, SAC).
    1. The only approaches for which this does not appear applicable are value-based methods such as DQN, Rainbow, or QT-OPT
* It is very simple to implement.
* The effectiveness of gSDE compared to other exploration noise structures is explored through rigorous and extensive experiments in simulation using a range of relevant algorithms (SAC, TD3, PPO, A2C) on multiple environments.
    1. The ablation studies (e.g. feature selection, noise re-sampling frequency) are relevant and illuminating.
    2. Figures 2 and 3 are especially interesting.
    3. All simulation results are reported with the mean and std error over 10 runs.
* The approach is further validated with the training of successful policies on three very different physical robots.
* The clarity of presentation and technical quality are excellent throughout.
    1. The explanations provided in both the main paper and appendix strike a good balance between detail and concision and I am confident that the results in the paper would be reproducible based on the information provided.

**Weaknesses**

* Low originality: gSDE is a minor modification to an existing method, SDE
* Lines 100 - 126: The claim that gSDE addresses all four of the identified limitations with SDE is not fully convincing
    1. (ii) Now the variance of the policy depends on the policy features dimension so needs to be tuned for each network.
    2. (iv) For similar reasons to those given in (iv) (lines 108 -109) shouldn’t the policy features now be normalized?
* Figure 2: Given the apparent linear relationship between the frequency of noise parameter re-sampling and continuity cost, the sudden drop-off to the Unstructured noise is surprising. It is achieving returns equivalent to gSDE-64 but with almost 2x normalized cost
    1. There is a similar puzzling relationship occurring in Figure 3b)
    2. To better understand this it would be helpful to add gSDE-4 and gSDE-2 to Figure 2 and gSDE 2 to Figure 3b).
    3. I would expect a somewhat smooth transition between gSDE-2 and unstructured noise. Regardless, adding these points would be illuminating.
* Grammar
    1. Line 13: “permits to have” → facilitates / enables?
    2. Line 54: “without the need of a” -> “without need of a” / “without the need for”
    3. Line 94: “allows to adapt” → “enables adaptation of”
    4. Line 184: “allows to have” → “permits a” / “makes it possible to”

**Comments and Questions**
* Lines 42 - 44: There is significant interest in and efforts made to apply deep RL to real robotic problems so it is not clear to me that a focus on simulation is the main reason for SDE not being combined with deep RL. Perhaps up to now other issues such as sample efficiency and generalizing to less structured environments or multiple tasks have occupied the attention of the community.
* Line 66: When actions are discrete exploration is also typically done in the action space.
* Table 2: Is there just one evaluation trajectory?
* It would be interesting to see the learning curves for the tendon-driven task
* Walking task: It would be interesting to compare the learning curve (Figure 5b) with SAC trained with a low pass filter on this task
* Driving task: It seems that the standard SAC exploration would not break this type of robot. It would be very interesting to see a comparison SAC, SAC w/ low pass filter, SAC + gSDE


**Summary Of Recommendation:**

This work is highly relevant to robotic learning using model free deep RL and the effectiveness of gSDE is convincingly demonstrated. Additionally the method is very simple and easy to implement. Experiments are well selected and thorough. The paper is lucid, easy to understand, and overall a pleasure to read. To further strengthen the paper I raised a few minor issues and questions.

Comment on impact 4:
The paper convincingly demonstrates the effectiveness of existing ideas on reducing shakiness of deep RL policies. The simplicity of the idea and ease of implementation increase the potential impact of this work.

---

> ### Author Response · Authors · 2021-08-26
> **Reply to reviewer 8waC**
>
>
> Dear Reviewer,
>
> Thank you very much for your time and valuable suggestions!
> Please find below our replies to your concerns and suggestions.
>
> >Lines 100 - 126: The claim that gSDE addresses all four of the identified limitations with SDE is not fully convincing
>
> >(ii) Now the variance of the policy depends on the policy features dimension so needs to be tuned for each network.
> >(iv) For similar reasons to those given in (iv) (lines 108 -109) shouldn’t the policy features now be normalized?
>
> It is true that the variance now depends on the policy features dimension. However, as shown in the experiments, one network architecture can work on a wide variety of tasks. Additionally, on all experiments conducted so far, because of the initialization of the network (small weights sampled around zero) and then of the adaptation of the noise during training, we didn't experienced any issue due to normalization (i.e., we did not had to tune the magnitude of the noise).
>
> Finally, using features from the last layer reduces drastically the number of parameters and operations in some cases (cf. example with image input in the reply to Reviewer #uHC8).
>
> >Figure 2: Given the apparent linear relationship between the frequency of noise parameter re-sampling and continuity cost, the sudden drop-off to the Unstructured noise is surprising. It is achieving returns equivalent to gSDE-64 but with almost 2x normalized cost
> > There is a similar puzzling relationship occurring in Figure 3b)
> > To better understand this it would be helpful to add gSDE-4 and gSDE-2 to Figure 2 and gSDE 2 to Figure 3b).
> >I would expect a somewhat smooth transition between gSDE-2 and unstructured noise. Regardless, adding these points would be illuminating.
>
> Thanks for the suggestion, we added gSDE-2 and gSDE-4 experiments to Figure 2, 3b) and Table 1 (gSDE-4 was left out from Figure 2 to keep the figure clear).
>
> In both figures we can see that the performance peaks for gSDE-4 or gSDE-8 with a drop on both sides. This effect indicates that even environments where there are no harmful effects of high frequency noise having slightly more structured exploration (could imagine those as larger steps in a random walk, which cover more state-space) will help the agent to discover better solutions, while there is some tipping point where having data on very precisely timed actions becomes more important for performance.
>
> This behavior can also be seen in the additional experiment on the MountainCar problem (requested by reviewer #bar8) that was added in the appendix. One can also note that gSDE is robust to hyperparameter change as it works (and even outperforms unstructured exploration) for a wide range of value for the noise sampling interval.
>
> >Table 2: Is there just one evaluation trajectory?
>
> There is only one evaluation trajectory, but it contains ~20 target poses (+ transient desired poses) which therefore corresponds to ~20 episodes (only an excerpt is shown in Figure 4b for clarity).
>
> >Grammar
>
> Thanks, updated the paper to fix with your suggestions.
>
> EDIT: - we added the learning curve for the tendon-driven task in the appendix as suggested by Reviewers #8waC and #bar8

---

> > ### Comment · Reviewer_8waC · 2021-09-02
> > **Response to authors**
> >
> > Dear authors,
> >
> > Thank you for responding to all of my comments and for your efforts to update the paper. I especially appreciate the additions of the gSDE-2 and gSDE-4 experiments as well as the MountainCar experiments included in the appendix.
> >
> > My recommendation remains a strong accept.
> >
> > This work is a valuable contribution to the research community since it convincingly shows that gSDE works on a wide range of problems and algorithms both in simulation and the real world. Given gSDE’s simplicity and quite general applicability I can imagine that if published, gSDE would be widely applied in practice.

---

### Official Review · Reviewer_bar8 · 2021-07-24

**Originality:** Good
**Technical Quality:** Fair
**Clarity Of Presentation:** Very Good
**Impact:** 3

**Recommendation:**

Weak Reject: I recommend rejecting the paper, but will not argue for my recommendation if the majority of other reviewers have a different opinion.

**Summary:**

The paper investigates how to achieve smooth exploration during deep reinforcement learning (deep RL), which is particularly an important problem when we want to apply deep RL to real-world environments. Basically, the author wants to improve unstructured exploration that shows noisy exploration at the early stage (Figure 1). The key idea is to improve the idea of State-Dependent Exploration (SDE) which defines action noises as a function of the state and an additional episodic parameter theta. The authors identified four disadvantages of SDE that limit exploration, and propose to 1) sample the episodic parameters every n steps instead of every n episodes and 2) use other feature values to generate noises.

The authors demonstrated that this approach can mitigate the known trade-off issue between safe exploration and learning performance. The authors also have shown that the proposed method can learn motor skills in three different real-world robotic environments, including a tendon-driven robot, a quadruped, and an RC car.


**Issues:**

First, I would suggest the authors discuss more scientific justification of gSDE.
I also suggest the authors conduct more extended experiments.


**Reviewer Expertise:**

Excellent: Expert knowledge on the topic of the paper

**Strengths And Weaknesses:**

+ The paper discusses a very important problem to deploy deep RL on real-world environments.
+ The paper demonstrates a good intellectual discussion about SDE, including its strengths and weaknesses.
+ The authors tested many interesting scenarios, including standard benchmark problems and real-robot learning.
- The design of gSDE sounds a little bit ad hoc.
- Some experiments are not complete: please refer to the text below.


**Summary Of Recommendation:**

In summary,my support is lukewarm. I like the problem itself and discussion about SDE, but gSDE seems to be a minor extension of SDE without much technical novelty.

First of all, I am very interested in real-robot learning, and this paper discusses one of the most important problems, how can we achieve safe exploration at the early stage of the learning process. And the paper describes pros and cons of SDE, which seem reasonable. I agree that SDE is based on a few design choices that can limit its exploration capability.

On the other hand, I’m not sure gSDE provides good solutions to it. The first idea is to resample the episodic randomization parameter theta in the middle of episodes, per two to 64 steps. I agree that they can be a good strategy for periodic locomotion tasks that are compared in Table I, but it cannot be a universal solution for general MDPs. I guess that this simple trick may not be very effective for other non-periodic (or episodic) tasks, such as manipulation. The example of a tendon-driven robot also seems a bit in favor of gSDE because the target trajectories have been periodically changed. I believe this can be a major drawback of the proposed approach.

The second innovation is to use the policy features, which are defined as the last layer values before the deterministic outputs. Yes, they may work better than the baselines. But I am a little bit concerned that the main reason for choosing this feature is simply because “we can in fact use any features”. I do not see any scientific conjectures here. What other features can we use? What if they do not work?

This lack of scientific justification requires the authors to provide more through experiments. Please see the below comments for more details.
- In general, I believe a much wider set of experiments is needed for supporting the paper’s statements.
- Figure 2 does not include SDE experiments. Is this intended? Not sure gSDE-Episodic is identical to SDE.
- I’m not sure Table 2 suggests that gSDE is the best. Low-pass filter achieves lower positional errors. Continuity cost does not say anything, if it is acceptable (= it is possible to learn a policy on real robots).
- Experiments on a quadrupedal robot and an RC car do not seem informative due to lack of statistics.

---

> ### Author Response · Authors · 2021-08-26
> **Reply to Reviewer bar8 (1/2)**
>
> Dear Reviewer,
>
> Thank you very much for taking the time to read our work thoroughly and your valuable advice!
> Please find below our replies to your concerns and suggestions.
>
> >The first idea is to resample the episodic randomization parameter theta in the middle of episodes, per two to 64 steps. I agree that they can be a good strategy for periodic locomotion tasks that are compared in Table I, but it cannot be a universal solution for general MDPs. I guess that this simple trick may not be very effective for other non-periodic (or episodic) tasks, such as manipulation.
>
> For exploration, there are two extremes: sampling the exploration parameter in every time-step (typical in action space exploration) and sampling  the exploration once at the beginning of the episode and keeping it fixed for the whole episode (often done in parameter space exploration, and in the original SDE).
> Note that the exploration itself is not constant in the action space in the second case as it gets transformed by the policy function, however is going to be consistent if a state gets visited multiple times and typically is changing smoothly over neighboring states.
> Structured exploration in action space, like Ornstein–Uhlenbeck achieves a similar effect.
> Both have their drawbacks: changing the exploration every time-step can damage the robot and might not even be effective as the system acts as a low-pass filter (l. 24), having consistent exploration for a whole episode can be inefficient as only limited information is gathered (l. 102).
> Resampling every n time-steps allows to trade off these properties, and also has been successfully been applied in the past in the discrete action case (hidden in the appendix of [1,2]).
>
> We believe this approach is applicable independently of whether the task is periodic or non-periodic.
> Its effectiveness depends on the relation of n to the number of total steps in the task.
> As both the value function/policy and the exploration are state dependent, and as the learned policy is not restricted to keeping constant for n time-steps, we believe that this does not prevent solving a general MDP.
> Note that we are not learning a feed-forward policy, like sometimes done in policy search for robotic tasks.
> For our experiments and in many tasks, the episode is longer than the time required to complete the task (typically with a reward term encouraging quick completion), even in the non-periodic setting.
> Hence, while the agent is searching for strategies to reach the goal, it is beneficial to explore different strategies during the same episode, i.e., change the exploration.
> The real robot task of controlling the neck is non-episodic, see details below.
>
> To backup our claims, we added an experiment in the appendix on a non-periodic task (MountainCar),
> a problem that cannot be solved by the original SAC (with unstructured noise) without additional external noise
> Because of the unstructured exploration, the commanded power oscillates at high frequency, making the velocity stay around the initial value of zero. The policy thus converges to a local minimum of doing nothing, which minimizes the consumed energy.
>
> On the other hand, SAC with gSDE works for a wide range of the noise sampling interval n (gSDE-4 to gSDE-128), while also improving a lot on the continuity cost at train time.
> If the sampling interval is too large (for instance with gSDE-Episodic or the original SDE), the agent would not explore enough during long episodes and then converge to the local minimum.
>
> [1] Meire Fortunato, et al. "Noisy Networks For Exploration." International Conference on Learning Representations.
> [2] Plappert, Matthias et al. "Parameter space noise for exploration". International Conference on Learning Representations.
>
> >The example of a tendon-driven robot also seems a bit in favor of gSDE because the target trajectories have been periodically changed. I believe this can be a major drawback of the proposed approach.
>
> There seems to be a misunderstanding regarding the tendon-driven robot task.
> The learning is episodic (only one target pose per episode) but the time between two episodes is short, which makes it look like periodic change in the video.
> The test trajectory shown in Figure 4b) (with periodic change to the target pose) is solely used during test time and not during training (where gSDE exploration is active).
>
> Additionally, the RC car task is not periodic.

---

> > ### Author Response · Authors · 2021-08-26
> > **Reply to reviewer bar8 (2/2)**
> >
> >
> > >But I am a little bit concerned that the main reason for choosing this feature is simply because “we can in fact use any features”. I do not see any scientific conjectures here. What other features can we use? What if they do not work?
> >
> > The reason for choosing the last layer is based on intuition and practical considerations.
> >
> > One can use the output of the different hidden layers of the policy as input features, which allows interpolating between the state (but yield issues raised in Section 3) to features close to the action.
> >
> > During the early development of the approach, we investigated using a separate network to provide features for the exploration function. At the end, the last layer features were the ones working on a good range of tasks without much tuning.
> >
> > Finally, using features from the last layer allows to be task-independent and reduces drastically the number of parameters and operations in some cases (cf. example with image input in the reply to Reviewer #uHC8).
> >
> > >In general, I believe a much wider set of experiments is needed for supporting the paper’s statements.
> >
> > Our paper already provides quantitative results on four simulated tasks with eight different algorithms, ablation study for two algorithms, one real robot experiment with three algorithms, and two additional real robot experiments.
> > The appendix contains additional experiments and ablation study on four simulated tasks with two additional algorithms (A2C, PPO).
> > We also now added an additional experiment on the MountainCar task.
> > Each experiment (leaving out real robot ones) is repeated 10 times.
> >
> > We would appreciate suggestions on which experiments to add.
> >
> > > Figure 2 does not include SDE experiments. Is this intended? Not sure gSDE-Episodic is identical to SDE.
> >
> > For SAC, as shown in Figure 3.a), gSDE-Episodic is very similar to SDE (but that's not the case for PPO).
> > We intentionally did not include SDE for this reason and to keep the Figure 2 clear, as we wanted to highlight the impact of repeating the noise on the compromise between smoothness and performance.
> >
> >
> > >I’m not sure Table 2 suggests that gSDE is the best. Low-pass filter achieves lower positional errors. Continuity cost does not say anything, if it is acceptable (= it is possible to learn a policy on real robots).
> >
> > If one is looking at the positional error only, it is true that using a low-pass filter allows to train a policy on a real robot that achieve a performance comparable to gSDE.
> >
> > However, one does not only care about positional error, as orientation and smoothness at test time are both important.
> > The continuity cost is for the trained agent and the significant difference in smoothness highlighted by the continuity cost can also be seen qualitatively seen in the test trajectory of Figure 4b).
> >
> > Apart from having more jitter, we also would like to pinpoint that action smoothing does not solve the root cause of the oscillations and is therefore not as principled as the proposed gSDE.
> > Also, this hand-crafted approach changes the problem (it hides the real action taken to the agent), adds one parameter that requires careful tuning, and introduces slowness: the policy cannot react immediately to a change in state.
> >
> > > Experiments on a quadrupedal robot and an RC car do not seem informative due to lack of statistics.
> >
> > To make the most use of our lab time (which is limited due to the Pandemic), we focused on simulation and on the tendon-driven robot to provide quantitative results and comparison between methods.
> > The additional real robot experiments are here to show the applicability of our method to a variety of real robots (which is in fact a concern of Reviewer #bar8).

---

> > > ### Comment · Reviewer_bar8 · 2021-08-26
> > > **Additional Comments**
> > >
> > > Dear authors,
> > >
> > > Thanks for the extended responses and additional experiments. They alleviated a lot of my primary concerns. I admit that the given set of experiments cover a good range of robot learning tasks.
> > >
> > > I still think whether gSDE is better than a low-pass filter I used to adopt. I learned that a low-pass filter could work on many different applications by tuning its hyperparameters. To answer this question, we may need thorough investigations by many researchers. But it should be good to have an alternative option, at least. In that sense, I am fine with the current version of Figure 4 because it is a good data point to be shared within the robotic community. Can you quickly compare the episodic rewards of a low pass filter and gSDE to Table 2?

---

> > > > ### Author Response · Authors · 2021-08-27
> > > > **Reply to additional comments**
> > > >
> > > > >Thanks for the extended responses and additional experiments. They alleviated a lot of my primary concerns. I admit that the given set of experiments cover a good range of robot learning tasks.
> > > >
> > > > Thank you very much.
> > > >
> > > > >Can you quickly compare the episodic rewards of a low pass filter and gSDE to Table 2?
> > > >
> > > > The episodic rewards are of the same magnitude for gSDE and the low-pass filter, as they achieve similar performance on the primary reward (reaching target) but not on the secondary (smoothness). We added the learning curves to the appendix.
> > > >
> > > > Regarding performance of the low-pass filter on the simulated environments, we did try in the past to make the filter work on them, but without much success... (careful tuning is probably needed)

---

### Author Response · Authors · 2021-08-26
**Reply to all reviewers**

We thank the reviewers for their feedback. We believe the quality of the paper will improve due to their observations and requests.

As noted by all reviewers, gSDE is an improvement of SDE that focuses on practical application (to have something that works in practice on a real robot), but as remarked by Reviewer #uHC8, "SDE is not used in RL, and gSDE solves the problems that make SDE less effective.".

Our goal is to propose an approach that solves practical issues, investigate the compromise between smoothness and performance usually overlooked, and to encourage interest in learning directly on real systems, where the current focus of DeepRL community is on learning in simulation.

## Changelog
- we updated Table 1, Figure 2 and 3b) to include gSDE-2 and gSDE-4 as requested by Reviewer #8waC
- we added an experiment in the appendix on a non-periodic task (Mountain Car) with different values of the noise sampling interval as requested by Reviewer #bar8
- we added a paragraph to clarify how gSDE saves parameters and computation when working with large state space such as images, as requested by Reviewer #uHC8
- we fixed the grammar issues spotted by Reviewer #8waC
- we added the learning curve for the tendon-driven task in the appendix as suggested by Reviewers #8waC and #bar8

---

### Meta-Review · Area_Chair_gCsT · 2021-08-13

**Recommendation:** Accept (Poster)
**Confidence:** 4

**Metareview:**

This paper proposes a simple and easy-to-implement idea for generating exploration noise for RL. This approach is especially making RL more effective when used on real-world robotic systems, i.e. it addresses the shaky behavior problem in practice. Though the proposed idea is just minor modifications to a existing method, i.e. SDE, and lacks of scientific justification or has very little technical novelty, the reviewers believed it can have great potential applicability, e.g. any continuous control problem. Though it is evaluated on a good set of tasks: on 4 MuJoCo tasks in simulation and 3 real-world systems. However it's encouraged to provide more results to support the paper’s statements, e.g. additional statistical information of experiments on a quadrupedal robot and an RC car.

The authors have made a great response that have addressed most concerns. Additional experiment results also support the paper's claims stronger. The proposed idea can be simple but it's very practically useful and worth being shared with the robotic community.

---

### Decision · Program_Chairs · 2021-09-13

**Decision:**

Accept (Poster)

**Comment:**

This paper proposes a simple and easy-to-implement idea for generating exploration noise for RL. This approach is especially making RL more effective when used on real-world robotic systems, i.e. it addresses the shaky behavior problem in practice. Though the proposed idea is just minor modifications to a existing method, i.e. SDE, and lacks of scientific justification or has very little technical novelty, the reviewers believed it can have great potential applicability, e.g. any continuous control problem. Though it is evaluated on a good set of tasks: on 4 MuJoCo tasks in simulation and 3 real-world systems. However it's encouraged to provide more results to support the paper’s statements, e.g. additional statistical information of experiments on a quadrupedal robot and an RC car.

The authors have made a great response that have addressed most concerns. Additional experiment results also support the paper's claims stronger. The proposed idea can be simple but it's very practically useful and worth being shared with the robotic community.